Genome-wide identification and expression analysis of fibrillin (FBN) gene family in tomato (Solanum lycopersicum L.)

Sun Huiru 1 2 shrpiaoliu@163.com
Ren Min 1
Zhang Jianing 1
1 College of Life Sciences, Yan’an University , Yan’an, Shaanxi Province , China
2 Shaanxi Key Laboratory of Chinese Jujube, Yan’an University , Yan’an, Shaanxi Province , China
Wang Xiukang
Electronic publication date: 2022 May 9
Publication date: 2022
Volume: 10
Electronic Location ID: e13414
Received 2022 Mar 23; Accepted 2022 Apr 19
Copyright: © 2022 Sun et al.
Copyright year: 2022
Copyright holder: Sun et al.
License: This is an open access article distributed under the terms of the Creative Commons Attribution License, which permits unrestricted use, distribution, reproduction and adaptation in any medium and for any purpose provided that it is properly attributed. For attribution, the original author(s), title, publication source (PeerJ) and either DOI or URL of the article must be cited.
License URL: https://creativecommons.org/licenses/by/4.0/

Keywords: Tomato, FBN genes family, Expression analysis, Leaf, Fruit

Funding: Specialized Research Fund for the Doctoral Program of Yan’an University YDBK2019-42 Natural Science Basic Research Plan of Shaanxi Province, China 2022JQ-159 Special Scientific Research Project of Education Department of Shaanxi Province, China 21JK0993 This work was supported by the Specialized Research Fund for the Doctoral Program of Yan’an University (No. YDBK2019-42), the Natural Science Basic Research Plan of Shaanxi Province, China (No. 2022JQ-159); and the Special Scientific Research Project of Education Department of Shaanxi Province, China (No. 21JK0993). The funders had no role in study design, data collection and analysis, decision to publish, or preparation of the manuscript.

==============================
Background

Fibrillin (FBN) proteins are widely distributed in the photosynthetic organs. The members of FBN gene family play important roles in plant growth and development, and response to hormone and stresses. Tomato is a vegetable crop with significantly economic value and model plant commonly used in research. However, the FBN family has not been systematical studied in tomato.

Methods

In this study, 14 FBN genes were identified in tomato genome by Pfam and Hmmer 3.0 software. ExPASy, MEGA 6.0, MEME, GSDS, TBtools, PlantCARE and so on were used for physical and chemical properties analysis, phylogenetic analysis, gene structure and conserved motifs analysis, collinearity analysis and cis-acting element analysis of FBN family genes in tomato. Expression characteristics of SlFBNs in different tissues, fruit shape near isogenic lines (NILs), Pst DC3000 and ABA treatments were analyzed based on transcriptome data and quantitative Real-time qPCR (qRT-PCR) analysis.

Results

The SlFBN family was divided into 11 subgroups. There were 8 FBN homologous gene pairs between tomato and Arabidopsis. All the members of SlFBN family contained PAP conserved domain, but their gene structure and conserved motifs showed apparent differences. The cis-acting elements of light and hormone (especially ethylene, methyl jasmonate (MeJA) and abscisic acid (ABA)) were widely distributed in the SlFBN promoter regions. The expression analysis found that most of SlFBNs were predominantly expressed in leaves of Heinz and S. pimpinellifolium LA1589, and showed higher expressions in mature or senescent leaves than in young leaves. Expression analysis of different tissues and fruit shape NILs indicated SlFBN1, SlFBN2b and SlFBN7a might play important roles during tomato fruit differentiation. All of the SlFBNs responded to Pst DC3000 and ABA treatments. The results of this study contribute to exploring the functions and molecular mechanisms of SlFBNs in leaf development, fruit differentiation, stress and hormone responses.

Introduction

Plastoglobules (PGs) are lipoprotein particles in plastid, which are involved in plant growth and development, and stress resistance. Fibrillin (FBN) proteins are plastid lipid-associated and highly conserved, which encoded by nuclear genes and are the most abundant in chloroplast PGs (Singh & McNellis, 2011). FBNs showed important regulatory roles in plastid stability, plant growth and development, and stress response (Rey et al., 2010; Simkin et al., 2007; Youssef et al., 2010; Ytterberg, Peltier & van Wijk, 2006).

FBN proteins contain a conserved plastid-lipids associated protein (PAP) domain and are widely exist in photosynthetic organisms from cyanobacteria to plants (Cunningham et al., 2010; Lohscheider & Bártulos, 2016; Simkin et al., 2007). FBN proteins were early found in the chromoplasts of red rose (Rosa rugosa) (Wuttke, 1976) and bell pepper (Capsicum annuum) fruit (Deruère et al., 1994). Subsequently, the similar proteins were isolated from thylakoids of potato (Solanum tuberosum) leaves (Pruvot et al., 1996), chromoplast in cucumber (Cucumis sativus) petals (Vishnevetsky et al., 1996), chloroplast PG of Arabidopsis thaliana leaves (Vidi et al., 2006) etc, which have been named PGL, PAP and FBN. Ultimately, these similar proteins are collectively referred to as FBN (Singh & McNellis, 2011). So far, the FBN families in different species have been divided into 12 groups (FBN1–FBN12) (Kim et al., 2018). The FBN12 group is only present in lower algal fungi (Lohscheider & Bártulos, 2016). Besides the unique PAP domain, FBN11 group also contain a protein kinase C domain (PKC), which indicates that members of this group might have other functions that need to be studied other than lipoprotein-related functions (Li et al., 2020). Moreover, the isoelectric point and molecular weight ranges of FBN family are wide, and they are distributed in different plastids, including chloroplasts, elaioplasts, chromoplasts and etioplasts, which might be related to the diversified functions (Kim et al., 2018).

The FBN family genes have showed important roles in various processes, such as plastid structure stabilization, organ development, stress response and hormone signal transmission (Kim, Lee & Kim, 2015). FBN1 protein was detected in the chromoplasts of ripe pepper fruits, and overexpression of this gene promoted the increase and aggregation of PGs in chromoplasts (Jotham et al., 2006). In addition, GUS activity of FBN1 promoter increased with tomato (Solanum lycopersicum) leaf aging (Georg et al., 2001). The expression of Pap2 (FBN1b) in Brassica rapa decreases with the aging of leaves (Kim, Wu & Huang, 2001). The content of FBN1 protein in bell pepper gradually increased with fruit ripening and reached the peak at the full fruit ripening stage. C40.4 (FBN1) in potato was expressed in leaves and multiple flower organs, and inhibition of this gene expression leaded to plant growth retardation and smaller tuber size (Monte, Ludevid & Prat, 2010). Some FBN family genes in rice (Oryza sativa) were respond to ABA and extreme temperature treatments (Lee et al., 2007; Li et al., 2020). CsaFBN1, CsaFBN6 and CsaFBN11 in cucumber were induced and up-regulated expressed under high-light and low temperature stress (Kim et al., 2018). The RNAi of LeChrC (FBN1) and mutants of fbi4 in apple (Malus domestica) and Arabidopsis were more sensitive to Botrytis cinerea, Erwinia amylovora and Pseudomonas syringae, respectively (Leitner-Dagan et al., 2006; Singh et al., 2010). The fbn6 mutant of Arabidopsis showed resistance to cadmium and high-light stresses (Lee et al., 2020). The expression of jasmonic acid (JA) synthesis related genes in fbn5 mutants were inhibited under high-light stress (Otsubo et al., 2018). Under drought stress, FBN1 showed significantly decreased expression in tomato flacca mutant with deficient in ABA synthesis (Gillet et al., 1998). FBN1 and FBN2 responded to light and cold stresses through JA synthesis pathway (Youssef et al., 2010). Some FBN family members in wheat (Triticum aestivum) were involved in the response to drought, cold, heat and stripe rust stresses (Jiang et al., 2020).

Tomato has rich nutritional value and is widely cultivated in the world as an important vegetable crop (Albacete et al., 2008). According to the Food and Agriculture Organization of the United Nations (FAO) statistics, the tomato cultivated area in the world has exceeded five million hectares, and the production has reached 187 million tons in 2020 (http://www.fao.org/faostat/en/#data/QC). The FBN family genes play key regulatory roles in many biological processes such as plant development and stress response. However, the vast majority of FBN family genes in tomato are unknown, and no comprehensive analysis of this family in tomato has been reported. In this study, bioinformatics methods were used to identify and analyze the phylogenetic relationship, conserved domain, collinearity, cis-acting element in promoter regions of SlFBN gene family. The expression characteristics of SlFBNs in different tissues, Pst DC3000 and ABA treatments were analyzed using transcriptome data and qRT-PCR, which provided a theoretical foundation for exploring the potential functions of SlFBNs.

Materials and Methods

Tomato material and treatments

The tomato used in this study was Micro-Tom. The tomato seedings grew under normal temperature conditions (26 °C/16 h in light condition and 18 °C/8 h in dark condition). When the seedings grew to 6-leaf stage, the young leaves, mature leaves and aging leaves (the yellowing part less than 5% of the whole leaf) were collected. The 6-leaf seedings with similar growth were treated with 100 μM ABA, and water as a control group. The leaves were collected at 0, 6, 12 and 24 h after treatments. The samples were immediately frozen in liquid nitrogen and stored at −80 °C. All samples were tested with tree independent biological replicates.

Identification of FBNs in tomato

Genomic data of tomato was downloaded from Sol Genomics Network (SL4.0, http://solgenomics.net/). The genomes of Arabidopsis, rice, maize (Zea mays), sorghum (Sorghum bicolor), cucumber and pepper were downloaded from TAIR (http://www.arabidopsis.org/), phytozome (https://phytozome.net/) and PGP (https://db.cngb.org/search/assembly/GCF_000710875.1/) databases, respectively. The PAP domain model of FBN family was obtained from Pfam database (http://pfam.xfam.org). FBN genes in tomato genome were screened by HMMER 3.0 based on the model (the threshold set as E < 1e−4). The normal mode of SMART (http://smart.embl-heidelberg.de/) and CD–search (https://www.ncbi.nlm.nih.gov/Structure/cdd/wrpsb.cgi) were used to further verify SlFBN family genes. Finally, members of FBN family in tomato were determined. The lengths, molecular weights(Mws), isoelectric points (pIs) and hydrophilicities of SlFBN proteins were predicted by ExPASy website (http://web.expasy.org/protparam/) (Artimo et al., 2012). The subcellular localizations of tomato FBN proteins were predicated using WoLF PSORT (https://wolfpsort.hgc.jp/) (Paul et al., 2007).

Phylogenetic tree, gene structure and conserved domain analysis

The FBN protein sequences from tomato, Arabidopsis, rice, maize, sorghum, cucumber and pepper were aligned and constructed phylogenetic tree using MEGA 6.0 by the neighbor-joining (NJ) with the p-distance model, and bootstrap was set to 1,000 replications.

The gene structures of SlFBNs were drawn in GSDS 2.0 (http://gsds.cbi.pku.edu.cn/) based on the introns-exon position information (Hu et al., 2015). The PAP domains of SlFBN proteins were analyzed using SMART website. The conserved motifs of SlFBN proteins were identified by MEME website (https://meme-suite.org/meme/). The maximum number of motif findings was set to 10, and other parameters were set to default values (Bailey et al., 2009).

Chromosome location and collinearity analysis

The chromosome distribution of SlFBNs was drawn by TBtools according to the location information from tomato genome database (Chen et al., 2020). MCScanx software was used to analyze the collinearity of FBN genes among Arabidopsis, rice and tomato (Wang et al., 2012), and then the collinearity diagram was shown through the Basic Circos module of TBtools.

Cis-acting elements in FBN promoter regions

The 1.5 kb sequences located upstream of the start codon of tomato FBN family genes were extracted to analyze the cis-acting elements on the PlantCARE website (http://bioinformatics.psb.ugent.be/webtools/plantcare/html/) (Lescot et al., 2002).

Transcriptome analysis of SlFBNs in different tissues and under Pst DC3000 treatment

The transcriptome data of SlFBNs in different tissues of Heinz and S. pimpinellifolium LA1589, under Pst DC3000 treatment in tomato varieties with different resistances (RG-PtoR: resistant, RG-prf3: sensitive and RG-prf9: sensitive) and in flower meristems at different developmental stages of LA1589 and three fruit shape near isogenic lines (NILs) in LA1589 background (WT, sun, ovate and fs8.1) (Wang et al., 2019) were obtained from TFGD (http://ted.bti.cornell.edu/cgi-bin/TFGD/digital/home.cgi). The expression heat maps of SlFBNs were drawn using TBtools.

RNA extraction and quantitative real-time PCR (qRT-PCR)

Total RNA was extracted from the collected samples using the Total RNA Kit (TIANGEN, Beijing, China). The cDNA was synthesized from 1 μg using the StarScript II First-strand cDNA Synthesis Mix kit (GenStar, Beijing, China) according to the manufacturer’s instructions. Then qRT-PCR was carried out with an Applied Biosystems StepOnePlus using RealStar Green Fast Mixture with ROX (2×) (GenStar, Beijing, China). The gene-specific primers used for qRT-PCR were designed by Primer 5.0 (Table S1). The qRT-PCR was performed as follows: step 1: 95 °C for 2 min; step 2: 40 cycles of 95 °C for 15 s, 60 °C for 30 s; and step 3: melting curve analysis. Three biological replicates and the 2−ΔΔCT method were used to calculate the relative expression level (Livak & Schmittgen, 2001). The tomato ACTIN gene (Solyc04g011500.3.1) was used as internal reference gene (Liu et al., 2022). The t-test was used to analyze the significance of the difference.

Results

Identification of FBN members in tomato

In our study, a total of 14 putative FBN genes were identified in tomato genome through screening using PAP domain (PF04755) and validating in SMART database. According to the homology with FBNs in Arabidopsis, SlFBNs were named as SlFBN1–SlFBN11 (Table S2). The physical and chemical properties of SlFBN proteins were analyzed. The result showed that the lengths of SlFBN proteins ranged from 206 amino acids (aa) (SlFBN9) to 541 aa (SlFBN11). The predicated Mw and pI ranged from 23.43 kDa (SlFBN9)–61.35 kDa (SlFBN11), and 4.65 (SlFBN2b)–9.72 (SlFBN3a). The hydrophobicities of SlFBNs were less than zero, indicating that they were all hydrophilic proteins. Most (11/14) of SlFBNs were located in chloroplasts, except for SlFBN3 (nucleus), SlFBN11 (nucleus) and SlFBN9 (mitochondria) by subcellar location predication.

Phylogenetic analysis of SlFBNs

To detect the evolutionary relationships of SlFBN family, total 85 FBN proteins were collected from rice (10), maize (13), sorghum (11), Arabidopsis (14), cucumber (10), pepper (13) and tomato (14) to construct the phylogenetic tree (Fig. 1). These FBN proteins were divided into 11 groups (Group 1–Group 11) which distributed FBN members from mono-and dicotyledons suggesting that the differences of FBN groups were completed before the separation of mono-and dicotyledons. In some species, group 1, 2, 3 and 7 were amplified. FBN members of tomato and pepper, which belong to Solanaceae, were firstly cluster in the same branch, while the FBN members from mono- and dicotyledons were clustered distantly. These results suggested that FBN family members in mono- and dicotyledons showed different evolutionary characteristics.

Figure 1 Phylogenetic tree of FBN family.

Chromosomal location and collinearity analysis of FBN genes

According to analysis of the chromosome positions, the 14 tomato FBN genes were unevenly distributed on seven chromosomes (Chr.01, Chr.02, Chr.03, Chr.08, Chr.09, Chr.10 and Chr.11). Among them, four SlFBNs were located on Chr.08, and 1 to 2 SlFBNs were located on the other six chromosomes (Fig. 2). The analysis of gene duplication event showed that there was no FBN gene duplication in tomato. To understand the collinearity relationship of FBN genes, the FBN homologous gene pairs between tomato and other plant species (Arabidopsis and rice) were found. There were eight collinear gene pairs between six SlFBNs and seven AtFBNs. SlFBN1 and SlFBN7b were collinear with two AtFBN genes (AtFBN 1a–AtFBN1b and AtFBN 7a–AtFBN7b), respectively. AtFBN2 was collinear with two SlFBN genes (SlFBN2a–SlFBN2b). SlFBN4 and SlFBN5 were collinear with one AtFBN gene, respectively (Fig. 3 and Table S3). There were no FBN homologous gene pairs between tomato and rice.

Figure 2 Chromosome location of SlFBN family genes.

Figure 3 Collinear analysis of FBNs in tomato and Arabidopsis thaliana.

The orange lines highlight the syntenic FBN gene pairs.

Gene structure and conserved domain of SlFBNs

To further explore the conservation and diversification of SlFBNs, the exon-intron structures and conserved motifs were analyzed. The gene structure analysis showed that the intron numbers of SlFBNs were various, ranging from 2 to 12 (Fig. 4A). Among them, SlFBN1, SlFBN2a, SlFBN2b, SlFBN9 contain two introns, which might be due to intron loss. SlFBN10 and SlFBN11 contained the 10 and 12 introns, respectively, which might be related to intron increase. Similar results were also found in FBN family genes in rice and Arabidopsis (Li et al., 2020).

Figure 4 Genetic structure of SlFBNs (A), conserved domains (B) and conserved motifs (C) of SlFBNs.

Analysis of the conserved domains of SlFBN family proteins revealed that all SlFBNs contained the PAP domain, which was unique to this family (Fig. 4B). SlFBN11 also contained a protein kinase C domain (PKC) besides PAP domain.

The 10 conserved motifs of SlFBN family members were analyzed to further analyze the characteristics of this family proteins. The sequences of the 10 conserved motifs were listed in Table S4. The results showed that all SlFBN proteins contained Motif2. Motif1 and Motif3 existed in most of SlFBNs, and the rest motifs existed in only individual SlFBN members (Fig. 4C). SlFBN11 only contained Motif2, which indicated that SlFBN11 might show different functions from other group members, combined with the analysis of gene structure and conserved domain.

Cis-element analysis of SlFBN promoter regions

The cis-acting element in promoter region can partly reflect the characteristic of gene expression. The cis-acting elements in the promoter regions of SlFBNs were analyzed and the result showed that all of the SlFBN promoters contained two to 15 light response elements (Fig. 5), which indicated that the functions of SlFBNs might be related with light response liking photosynthesis. The hormone responsive elements, especially ethylene, MeJA and ABA, were widely distributed in the SlFBN promotes. The promoters of SlFBN10, SlFBN6 and SlFBN4 contained 11, nine and five ABA response elements (ABRE), respectively, suggesting that these genes might be directly or indirectly involved in ABA response pathway. In addition, auxin, gibberellin and salicylic acid response elements were distributed in some SlFBN promoters. Drought response element (MBS), pathogen response element (W-box), trauma response element (WUN-Motif) and defense and stress response element (TC-rich repeats) were distributed in the 6, 5, 3 and 3 SlFBN gene promoters, respectively. In addition to the above elements, individual SlFBN promoters also distributed endosperm expression elements (GCN4-Motif), meristematic expression regulation elements (CAT-box) and circadian control elements (circadian).

Figure 5 Cis-acting element distributions of SlFBN family gene promoters.

Expression patterns of SlFBNs in different tissues

In order to explore the biological functions of SlFBNs in tomato growth and development, the expression patterns of SlFBNs in different tissues of tomato cultivar Heinz (Fig. 6A) and wild species S. pimpinellifolium LA1589 (Fig. 6B) were analyzed using published transcriptome data. Similar expression characteristics, which was that most of SlFBNs were preferentially expressed in leaves, were observed in Heinz and LA1589. Meanwhile, In Heinz, four SlFBNs (SlFBN1, SlFBN2b, SlFBN3a and SlFBN7a) were highly expressed in flowers. In LA1589, except SlFBN2b, the other three SlFBNs were also highly expressed in flowers. In addition, the expressions of some SlFBNs (SlFBN4, SlFBN6, SlFBN8, SlFBN9 and SlFBN11) were increased with fruit ripening both in Heinz and LA1589, suggesting that these genes might be involved in regulating tomato fruit ripening.

Figure 6 Expressions of SlFBNs in different organs.

(A) Expression profile of SlFBNs in different organs of cultivated tomato cultivar Heniz; (B) expression profile of SlFBNs in different organs of wild species S. pimpinellifolium LA1589; (C) expression profile of SlFBNs in LA1589, sun, ovate and fs8.1. The three red asterisks represent the differentially expressed genes between LA1589 and the three fruit-shape NILs (sun, ovate and fs8.1). Scale bars in A, B and C represent log2-transformed FPKM values.

The SlFBN expressions in flower meristems at 4, 6, 8, 10, 13 and 16 days post-initiation of floral meristem (DPI) of LA1589 and three fruit shape NILs (sun, ovate and fs8.1) were analyzed to further analyze their expression characteristics at different flower development stages (Fig. 6C). It was found that there was no significant difference in the expressions of 11 SlFBNs at different flower meristem stages. Notably, SlFBN1, SlFBN7a and SlFBN2b with high expressional levels in Heinz and LA1589 flower showed significantly higher expressions in three NILs than wild type at 16 DPI. The result indicated that SlFBN1, SlFBN7a and SlFBN2b might play important roles in tomato fruit early differentiation.

Tissue expression analysis showed that SlFBN family genes were mostly expressed preferentially in leaves. The expressions of SlFBNs at different development stages of tomato leaf were analyzed using qRT-PCR to explore the possible functions in tomato leaf development (Fig. 7 and Table S5). The expressions of 11 SlFBNs (except SlFBN2b, SlFBN7a and SlFBN7b) in young leaves were significantly different from those in mature or senescent leaves. These SlFBNs showed up-regulated expressions with leaf development except SlFBN11, which showed opposite trend. The results indicated that SlFBN family genes generally perform functions during leaf development. The study of FBN1 in bell pepper and tomato confirmed this result (Deruère et al., 1994).

Figure 7 Expressions of SlFBNs at different development stages of tomato leaf.

The expression levels of SlFBNs were tested by qRT-PCR and estimated by the 2−ΔΔCT method. Leaf-1, Leaf-2 and Leaf-3 represent young, mature and senescent leaves, respectively; The error bars show the standard error (SE) of three biological replicates. The p value was calculated through t-test. Asterisk indicate the significant difference compared with control (Leaf-1). * and ** indicate p < 0.05 and p < 0.01, respectively.

Expression profiles of SlFBNs under Pst DC3000 treatment

Previous studies showed that FBN family genes played important regulatory roles in stress response. The published transcriptomic data were used to analyze the expression characteristics of SlFBN family genes in tomato varieties with different resistances (RG-PtoR: resistant, RG-prf3: sensitive and RG-prf9: sensitive) under Pst DC3000 treatment (Fig. 8). The result showed that all of SlFBN family genes could respond to Pst DC3000 treatment, and 12 SlFBNs showed higher expression levels in resistant varieties than in sensitive varieties. The expressions of SlFBN1 and SlFBN11 in resistant varieties were higher than in sensitive varieties at 4 h, while the opposite trends were observed at 6 h under Pst DC3000 treatment.

Figure 8 Expressions of SlFBNs under Pst DC3000 treatment.

The scale bar represents log2-transformed FPKM values.

Expression profiles of SlFBNs under ABA treatment

The ABA response elements were generally distributed in SlFBN promoter regions. The expression levels of SlFBNs under ABA treatment were analyzed by qRT-PCR (Fig. 9 and Table S6). It was found that all of SlFBNs had significantly differences compared to the control. Compared to the 0 h, the expressions of SlFBN1, SlFBN2a, SlFBN2b, SlFBN3a and SlFBN5 were up-regulated, and the expressions of SlFBN3b, SlFBN4, SlFBN7b, SlFBN8 and SlFBN9 were down-regulated. The expression of SlFBN7a and SlFBN10 were increased firstly and then decreased. SlFBN6 and SlFBN11 showed early down-regulation followed by up-regulation. Notably, SlFBN11 showed the most significant response to ABA, and its expression increased 27.0 times at 12 h and 9.7 times at 24 h compared with the 0 h, suggesting that this gene was likely to involved in the ABA signal pathway.

Figure 9 Expressions of SlFBNs under ABA treatment.

The expression levels of SlFBNs were tested by qRT-PCR and estimated by the 2−ΔΔCT method. 0, 6, 12 and 24 h represent 0, 6, 12 and 24 h under ABA treatment, respectively; The error bars show the standard error (SE) of three biological replicates. The p value was calculated through t-test. Asterisk indicate the significant difference compared with control (0 h). Asterisks (* and **) indicate p < 0.05 and p < 0.01, respectively.

Discussion

FBN family proteins were early discovered in fibrils of chromoplasts (Deruère et al., 1994) and were involved in the various plant tissues growth and development, stress and hormone signal response (Jiang et al., 2020; Liu et al., 2018). In recent years, FBN gene families have been identified in several plants, such as rice (Li et al., 2020), cucumber (Kim et al., 2018) and wheat (Jiang et al., 2020). In this study, 14 FBN family genes were identified in tomato genome (Fig.1 and Table S2) and unevenly distributed in the chromosomes (Fig. 2). The similar numbers of FBN family were found in diploid plants such as rice (Li et al., 2020) and Arabidopsis (Singh & McNellis, 2011), but less than that in heterogenous hexaploid wheat, which might be related to the genome sizes and gene duplication events. Subcellular localization prediction found that most (11/14) of SlFBNs were located in chloroplasts (Table S2), and similar results were found in the studies of FBN family in wheat (Jiang et al., 2020) and rice (Li et al., 2020), suggesting that FBN family gene might be involved in controlling chloroplast structure. Phylogenetic analysis showed that SlFBN family genes were divided into 11 groups (Fig. 1). SlFBN11 belonging to group11 were significantly different from other group members in gene structure and conserved domains, suggesting that the gene might undergo new functionalization.

The gene structures of SlFBNs were distant with 2 to 12 introns (Fig. 3A), but the members in the same group showed similar intron numbers and the conserved motif arrangement suggesting that they might have similar functions and the members in different groups have diverse functions, especially SlFBN11. The concordant results were found in cucumber (Kim et al., 2018) and rice (Li et al., 2020). The collinearity analysis showed that tomato FBN genes have no homologous gene pairs with rice FBN genes and eight homologous gene pairs with Arabidopsis FBN genes (Fig. 3) indicating that FBN genes in mono- and dicotyledonous species might be divergence during evolution. We analyzed the cis-acting elements of SlFBN promoters to explore the SlFBNs expression characteristics. In SlFBN gene family, light and hormone response elements (such as ethylene, MeJA and ABA) are widely distributed in the promoter regions (Fig. 5), and similar regulatory elements are also widely distributed in FBN family genes of wheat (Jiang et al., 2020), suggesting that this family genes might be involved in the regulatory pathways of light and hormone.

The tissue expression analysis was carried out to explore the biological functions of tomato FBN family genes, it was found that the majority of SlFBNs were preferentially expressed in leaves (Figs. 6A and 6B). Further analysis of the expressions of SlFBNs in different development stages of leaf showed that most (11/14) of SlFBNs showed significantly increased expression trends with leaf maturation or aging (Fig. 7). This result was consistent with the study of subcellular localization and the GUS activity characteristics of SlFBN1 in process of tomato leaf senescence (Georg et al., 2001), which suggested that SlFBN1 might be involved in leaf maturation or senescence by regulating chloroplast structure. Previous studies have found that FBN genes were involved in regulating the fruit development of bell pepper (Kilcrease et al., 2015), satsuma mandarin (Citrus unshiu Marc.) (Moriguchi et al., 1998), sweet orange (Citrus sinensis) (Muccilli et al., 2009) and so on. In our study, SlFBN1, SlFBN7a and SlFBN2b were highly expressed in flowers, and the expression levels in flower meristem of three fruit shape NILs (sun, ovate and fs8.1) at 16 DPI were higher than those in LA1589, and the most significant differences were in sun (Fig. 6C). These results indicated that SlFBN1, SlFBN7a and SlFBN2b might be involved in regulating tomato fruit early differentiation and might be related with sun, ovate or fs8.1 gene loci. The studies of LeChrC in tomato (Leitner-Dagan et al., 2006), FBI4b in apple and AtFBN4 in Arabidopsis (Singh et al., 2010) showed that FBN genes could respond to pathogen infection. Through analysis of the transcriptomic data, all of SlFBNs could respond to Pst DC3000 treatment and most (12/14) of them showed higher expressions in resistant varieties than in sensitive variety (Fig. 8), which verified the previous studies. Analysis of the response of SlFBN family genes to ABA treatment showed that all of FBNs in tomato showed significant response to ABA treatment, especially SlFBN11 (Fig. 9), suggesting that this gene might play certain roles in the ABA signal pathway. This result was different from the FBN family study in rice (Jiang et al., 2020) indicated that functional differentiation was occurred in FBN family between mono-and dicotyledons. Together, most of SlFBN family genes were involved in leaf development and all of them could respond to Pst DC3000 and ABA treatments. SlFBN1, SlFBN7a and SlFBN2b might play roles in regulating tomato fruit differentiation. In addition, SlFBN11 might show different functions from other SlFBNs.

Conclusions

We identified 14 FBN genes in tomato genome, which were divided into 11 groups and unevenly distributed on seven chromosomes. There were eight FBN homologous gene pairs between tomato and Arabidopsis and no homologous gene pairs between tomato and rice. The FBN gene structures were divergent. The analysis of cis-acting elements found that hormone responce elements were extensive discovered in SlFBN promoter regions. The results of expression analysis were found that SlFBN family genes might show certain functions in leaf development, fruit differentiation, stress and hormone responses. These results could provide relevant information for further study on the biological functions of FBN family genes.

Supplemental Information

Supplemental Information 1 Primers used in qRT-PCR analysis of SlFBNs.

Click here for additional data file.

Supplemental Information 2 Physical and chemical properties of FBN family in tomato.

Click here for additional data file.

Supplemental Information 3 Homologous FBN gene pairs between tomato and Arabidopsis.

Click here for additional data file.

Supplemental Information 4 The 10 conserved motifs of SlFBN family proteins.

Click here for additional data file.

Supplemental Information 5 Raw data of SlFBNs expressions at different development stages of tomato leaf for data analyses and preparation for Figure 7.

The expression levels of SlFBNs were tested by qRT-PCR and estimated by the 2−ΔΔCT method. Leaf-1, Leaf-2 and Leaf-3 represent young, mature and senescent leaves, respectively. The error bars show the standard error (SE) of three biological replicates. The p value was calculated through t-test. Asterisk indicate the significant difference compared with control (Leaf-1). * and ** indicate p＜0.05 and p＜0.01, respectively.

Click here for additional data file.

Supplemental Information 6 Raw data of SlFBNs expressions under ABA treatment for data analyses and preparation for Figure 9.

The expression levels of SlFBNs were tested by qRT-PCR and estimated by the 2−ΔΔCT method. 0, 6, 12 and 24 h represent 0, 6, 12 and 24 h under ABA treatment, respectively. The error bars show the standard error (SE) of three biological replicates. The p value was calculated through t-test. Asterisk indicate the significant difference compared with control (0 h). * and ** indicate p＜0.05 and p＜0.01, respectively.

Click here for additional data file.

Additional Information and Declarations

Competing Interests

Author Contributions

Data Availability

The authors declare that they have no competing interests.

Huiru Sun conceived and designed the experiments, analyzed the data, prepared figures and/or tables, authored or reviewed drafts of the paper, and approved the final draft.

Min Ren analyzed the data, authored or reviewed drafts of the paper, and approved the final draft.

Jianing Zhang performed the experiments, authored or reviewed drafts of the paper, and approved the final draft.

The following information was supplied regarding data availability:

The raw data are available in the Supplemental Files.

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
