# Peer review of "Genome-wide identification and expression analysis of fibrillin (FBN) gene family in tomato (Solanum lycopersicum L.)"

_PeerJ, doi:10.7717/peerj.13414_

## Round 0.1 · original submission · Minor Revisions

Please carefully check whether the figures and tables are referenced in the paper in accordance with the journal format requirements. The reference format should be checked carefully according to the journal requirements.

Reviewer 1 ·

Basic reporting

no comment

Experimental design

no comment

Validity of the findings

no comment

Additional comments

FBN play important roles in tomato development, and they also involved in response to phytohormone and different stresses. This paper mainly explore their function with qRT-PCR in differernt samples. There are some questions need to be answered.
(1 )In your paper, SlFBN1, SlFBN2b and SlFBN7a were highly expressed in flowers, but the three genes were located in chloroplast, and chloroplast content in flowers may not so high, please present your viewpoint.
(2) As known that FBN in tomato may originated from a common ancestor, there is no gene duplication among 14 SlFBNs, please give your opinion.
(3) SlFBNs have no collinearity with rice, and present collinearity with Arabidopsis, but rice has bigger genome size than Arabidopsis, please state your standpoint.

·

Basic reporting

no comment

Experimental design

no comment

Validity of the findings

no comment

Additional comments

several minor comments,
1. please add note of the legend in Fig.6 and Fig.8 what dose -1.5, 0 and 1.5 means?
2. P value should be p value.
3. several formatting errors in references, such as Line 408-410,"2020" appeared twice, Line365-367, "37(web server issue)"?

Reviewer 3 ·

Basic reporting

no comment

Experimental design

no comment

Validity of the findings

no comment

Additional comments

In this article, the authors identified 14 FBN genes in the genomes of tomato, which divided into 11 subgroups, studied the phylogenetical and collinearity relationship among the subfamilies of FBN from different species, and found 8 homologous gene pairs between tomato and Arabidopsis. Furthermore, the authors provided detailed expression data of SlFBNs in tomato through transcriptomic data analysis and qRT-PCR. The major interesting findings were that: 1. The cis-acting elements of hormone (especially ethylene, MeJA and ABA) were widely distributed in the SlFBN promoters. 2. most of SlFBNs were preferentially expressed in leaves and showed higher expression in mature or senescent leaves than in young leaves ". 3. SlFBN1, SlFBN7a and SlFBN2b with high expressional levels in flower showed significantly higher expressions in three NILs than wild type. 4. All of SlFBNs were responded to Pst DC3000 and ABA treatments. These results provide some useful information for the next step of exploring the functions and molecular mechanisms of SlFBNs in leaf development, fruit differentiation, stress and hormone responses. However, with some shortcomings, the present vision needs improvements for publication.
1. In materials & methods section: Line 116-117 provides the genomic data of tomato. But at present, there are different genome assembles of tomato. It is necessary to showed the version information of tomato genome.
2. In materials & methods section: The authors should provide the detailed information on biological duplications of tomato samples under different treatments.
3. In introduction section: Line 81 showed the fbi4 mutants in apple and Arabidopsis. The “apple” should provide the Latin name as first appearance.
4. In materials & methods section: in Identification of FBNs in tomato paragraph, line 123, the authors mentioned that using the SMART "to screened FBN genes in tomato genome", didn't mention which mode of SMART. Since two mode applied different algorithm, it may cause different results in analysis.
5. In materials & methods section: in RNA extraction and quantitative real-time PCR (qRT-PCR) paragraph, line 173, the authors mentioned that ACTIN gene was used as internal reference gene. There are many ACTIN genes in tomato genome. So, the authors should provide the gene accession No. of ACTIN used in this article.
6. In results section: in Cis-element analysis of SlFBN promoter regions paragraph, line 238-239, the authors found “The hormone responsive elements, especially ethylene, MeJA and ABA, were widely distributed in the SlFBN promotes.” by cis-acting element analysis. The authors need to explain why they choose ABA, when they analyzed the expression profiles of SlFNBs under plant hormones.
7. There are many format errors in references. For example, Line 404, the Latin name of tomato " Solanum lycopersicum " in the title " Hormonal changes in relation to biomass partitioning and shoot growth impairment in salinized tomato (Solanum lycopersicum L.) plants "need to be italic. The same questions are in Line 426, 435, 442, 473 and so on. The For the name of journals, in some journals the first letter of each word is capitalized (e.g., Nucleic Acids Research, Journal of Bacteriology), while in others, not all the same (e.g., Molecular plant, Journal of experimental botany, The Plant cell). Some journals are used full name (e.g., Plant Physiology, Plant Science.) while others are abbreviated (Plant Sci, Plant Physiol, et al.).
8. The extra blanks occurred multiple times in whole article.
9. The Figure 2 and 3: the names of chromosomes are different with Table 2. The authors should unify the chromosome names.
10. The Figure 4: the font of legends in figure 4-A and B is too small to read.
11. The legend of Figure 6: There no statement about the meaning that three red asterisks represent in Figure 6-C. The authors need to provide additional explanation.

---

## Round 0.2 · accepted · Accept

Please carefully check the language, subscript, italics, etc. Ensure that every detail meets the journal requirements.